# Effect of Ultraviolet Radiation on Reducing Airborne *Escherichia coli* Carried by Poultry Litter Particles

**DOI:** 10.3390/ani12223170

**Published:** 2022-11-16

**Authors:** Xuan Dung Nguyen, Yang Zhao, Jeffrey D. Evans, Jun Lin, Brynn Voy, Joseph L. Purswell

**Affiliations:** 1Department of Animal Science, The University of Tennessee, Knoxville, TN 37996, USA; 2Poultry Research Unit, Agriculture Research Service, United States Department of Agriculture (USDA), Mississippi State, MS 39762, USA

**Keywords:** airborne *E. coli*, barn-to-barn infection, dry aerosolization, poultry houses, ultraviolet radiation

## Abstract

**Simple Summary:**

Airborne *Escherichia coli* (*E. coli*) has been proven to be a threat to the poultry industry. This study aimed at examining the effect of UV light (with the wavelength of 254 nm) on the inactivation of airborne *E. coli* carried by poultry dust particles. A newly designed system was used to assess the inactivation rates of UV light. *E. coli* inactivation was tested at different contact times (from 5.62 to 0.23 s) and different UV irradiance levels (of 1707 µW cm^−2^ and 3422 µW cm^−2^). The airborne *E. coli* was reduced significantly for all treatments with UV lamps. The inactivation rates can reach over 99.87% and 99.95% at 0.11 ± 0.02 m s^−1^ wind speed with of 1707 µW cm^−2^ and 3422 µW cm^−2^. The results may provide an insightful understanding of the UV effect on airborne *E. coli*.

**Abstract:**

Airborne *Escherichia coli* (*E. coli*) originating in poultry houses can be transmitted outside poultry farms through the air, posing risks of barn-to-barn infection through airborne transmission. The objective of this study is to examine the effect of ultraviolet (UV) light on the inactivation of airborne *E. coli* carried by poultry dust particles under laboratory conditions. A system containing two chambers that were connected by a UV scrubber was designed in the study. In the upstream chamber of the system, airborne *E. coli* attached to dust particles were aerosolized by a dry aerosolization-based system. Two sets of air samplers were placed in the two chambers to collect the viable airborne *E. coli*. By comparing the concentration of airborne *E. coli* in the two chambers, the inactivation rates were calculated. The airborne *E. coli* inactivation rates were tested at different contact times with the aid of a vacuum pump (from 5.62 to 0.23 s of contact time) and different UV irradiance levels (of 1707 µW cm^−2^ and 3422 µW cm^−2^). The inactivation rates varied from over 99.87% and 99.95% at 5.62 s of contact time with 1707 µW cm^−2^ and 3422 µW cm^−2^ of UV irradiance to 72.90% and 86.60% at 0.23 s of contact time with 1707 µW cm^−2^ and 3422 µW cm^−2^ of UV irradiance. The designed system was able to create the average UV irradiation of 1707 µW cm^−2^ and 3422 µW cm^−2^ for one UV lamp and two UV lamps, respectively. The findings of this study may provide an understanding of the effect of UV light on the inactivation of airborne *E. coli* carried by dust particles and help to design an affordable mitigation system for poultry houses.

## 1. Introduction

The United States is a major producer of eggs and poultry meat worldwide. As a 35 billion dollar sector, the poultry industry provided approximately 1 million jobs for the U.S. in 2020 [1]. However, this sector of the economy is extremely vulnerable to infectious diseases brought on by pathogenic bacteria, such as Avian Pathogenic *Escherichia coli* (APEC). All ages of birds and all types of poultry houses could be affected by the APEC-caused diseases [2]. One of the key factors contributing to the financial losses of the global poultry sector was thought to be APEC [3]. These microorganisms are frequently found in the lower gastrointestinal tracts of chickens and other warm-blooded animals, as well as in the environment where the animals live. Hemorrhagic colitis, gastroenteritis, and urinary tract infections are among the intestinal symptoms (colibacillosis) brought on by APEC. The cost of poultry losses, mortalities, medical expenses, and decreased feed efficiency were the main causes of the economic losses caused by APEC [4]. According to a previous study [5], roughly 40% of broiler carcasses that were condemned included APEC, and 30% of broiler flocks in the United States had subclinical colibacillosis [6].

The air in poultry buildings contains not only smells and pollutants, but also a large number of pathogenic bacteria, such as *Escherichia coli* (*E. coli)*. Young chicks can get infected by vertical transmission from an infected ovary, oviduct, or contaminated eggs passing through the cloacal manures of infected or carrier hens. When birds are infected at a young age, they may have few minimal symptoms of sickness yet still become carriers. In older birds, infection of *E. coli* has a predisposition for reproductive organs, which frequently leads to infection of ovarian follicles and, as a result, transovarial transmission of the illness. Then, *E. coli* are carried out by poultry manure. The *E. coli* is first excreted onto poultry litter, then plowed up and dispersed into the air by bird activities [7]. Past studies reported that the concentration of airborne *E. coli* can be up to 4 log_10_ CFU m^−3^ in poultry houses [8]. After being aerosolized into the air, the airborne *E. coli* can migrate into the poultry house following the airflow of ventilation fans. Therefore, there is a high possibility that the birds in the poultry houses can receive the airborne *E. coli* through inhalation and contact with the areas where the airborne *E. coli* settled. The birds become sick by inhaling dust mixed with feces, which can carry up to 10^6^ CFU of *E. coli* per gram [9]. This aerogenic mode of infection is thought to be the primary cause of systemic colibacillosis or colisepticemia [10]. In addition, the airborne *E. coli* can be emitted outside the poultry houses, which poses risks to barn-to-barn airborne transmission [8]. The previous study mentioned that there was no significant difference in concentration airborne *E. coli* between the inside and downwind locations within 10 meters. A potential solution that may reduce airborne *E. coli* emitted outside the poultry houses at an affordable cost for farmers is necessary for mitigating the airborne transmission of *E. coli*.

Ultraviolet-C(UVC) which covers the wavelength range from 100–300 nm was well studied in the food industry and is known as a method that can inactivate microorganisms by inhibiting DNA replication [11]. The previous study [11] also reported that UVC light was very effective to disinfect *E. coli* in water, droplets, and surfaces in the food industry processing. Specifically, UV with a wavelength of 254 nm showed the highest performance in terms of disinfecting pathogens [11]. In poultry houses, the airborne *E. coli* can be carried by dust particles [12,13] that might prevent UV light exposure, and thus the dust particles can protect the airborne *E. coli* from being irradiated [14]. In addition, the variation of environmental conditions in poultry houses such as ventilation systems or air flows, temperature, and relative humidity (RH) lead to variable contact time and resistance to UV light. The inactivation efficiency in the poultry houses might be different from the food industry. Therefore, the inactivation efficiency of UV light on poultry litter-based airborne *E. coli* needs further investigation.

The objective of this study was to investigate the inactivation efficiency of UV light (wavelength of 254 nm) on airborne *E. coli* carried by poultry dust particles in laboratory conditions. The laboratory conditions remained stable at about 22.6 °C with an RH of 60%. A system that simulated the conditions of the poultry house was designed to evaluate the inactivation rate. The tested wind speeds were from 0.11 to 2.61 m s^−1^, corresponding to the contact time from 5.6 s to 0.23 s. In addition, the UV intensity, dust concentrations, and size distribution of *E. coli* carried by poultry dust particles were also recorded.

## 2. Materials and Methods

To evaluate the effect of UV light, the experiment was conducted in a Biosafety Level 2 (BSL-2) laboratory located at the Animal Science Department, University of Tennessee, Knoxville, TN 37996, U.S. The Institutional Biosafety Committee at The University of Tennessee has approved this study under the protocol IBC-21-572-2.

### 2.1. Microorganism and System Descriptions

#### 2.1.1. Preparation of *E. coli* Solution

The *E. coli* strain utilized in this investigation was ATCC^®^ 25922 (ATCC^®^ 25922), which was obtained from the American Type Culture Collection (ATCC, Manassas, VA, USA). *E. coli* strain was cultured at 37 °C, 150 rpm for 24 h in ATCC^®^ Medium 18 (Tryptic Soy Broth ‘TSB’ and Tryptic Soy Agar ‘TSA’, ATCC, Manassas, VA, USA). The bacterial concentrations of *E. coli* in the solution after 24 h were determined by the traditional serial dilution process [12,15]. The concentration was approximately 9 log_10_ colony-forming unit (log_10_ CFU) mL^−1^.

#### 2.1.2. Litter Preparation

The litter preparation was performed in the same way as in our previous studies [12,15]. Litter was taken from a commercial broiler farm. It was subsequently returned to the Biosafety Level 1 (BSL-1) laboratory for an analysis of the dry matter content (DMC). It was then autoclaved at 121 °C for 20 minutes before being separated into identical-size aluminum boxes weighing 6 kg each. The boxes were packed with aluminum foil and coated with plastic lids to prevent contamination. They were kept in a 4 °C fridge until they were utilized.

It was necessary to prepare the litter so that the bacteria were distributed evenly. This experiment required 240 g of litter, which was evenly distributed among 40 ceramic cups. In a prior experiment, the capacity to generate airborne dust was examined [12], and the findings revealed that 240 g of litter put to the mixer produced dust concentrations ranging from 0.9 to 1.1 mg m^−3^ which is within the average range of dust concentration in a commercial chicken farm [16]. To prepare litter inoculated with *E. coli*, a set of 43 ceramic cups (40 cups for experiment plus 3 controls) which were identical in shape was used to hold the litter. In each cup, 6 g of litter was prepared and mixed with 6 mL of *E. coli* cultured solution. The 6 mL bacteria solution was sprayed equally onto the litter in each cup. Meanwhile, a sterile metal spoon was used to gently mix the litter and *E. coli* solution. After that, the mixtures were dried for 48 hours at 20.8 °C and 40–65% RH till the DMC reached about 80% and was appropriate for aerosolization. The concentration of *E. coli* in the control cup was determined by adding TSB to the mixture until the total volume of each cup reached 15 mL. An automated pipette was then used to collect 1 mL of the solution in each cup. A conventional serial dilution approach was used to determine the culturable *E. coli* in the 1 mL solution. The concentration of *E. coli* in the cup was approximately 8 log_10_ CFU g^−1^ litter after the drying process. The *E. coli*-containing litter was then moved from 40 ceramic cups to the mixer’s metal bowl for aerosolization. Before aerosolization, the litter was gently mixed again in the bowl.

#### 2.1.3. Test Chambers

Two connected acrylic chambers were used in this study. The upstream chamber (2100 series, Cleatech, Orange, CA, USA) was a non-vacuum unit with two internal access doors with stainless steel frame, and a detachable completely gasketed rear wall. The dimension of the test chamber was 1.5 m L × 0.6 m W × 0.6 m H. The dimension of the downstream chamber (2200 series, Cleatech, Orange, CA, USA) was 0.7 m L × 0.6 m W × 0.6 m H. The two chambers were connected by an aluminum tube installed with two UV lamps. The scrubber helped the air inside of two chambers to be circulated. The dimension of the scrubber was 0.24 m D × 0.6 m H. The chambers were well sealed to prevent dust from spilling out. Temperature and RH sensors were equipped for continuously monitoring the inside thermal environment.

#### 2.1.4. Aerosolization System

In this study, a stand mixer (model DCSM350GBRD02, New York, NY, USA) was used to dry aerosolize airborne *E. coli*. The dimension of the mixer was 0.3 m L × 0.2 m W × 0.3 m H with a 3.3 L stainless steel bowl. It was operated at maximum speed to ensure the bacteria concentration in the air was high enough for the samplers were able to detect it. A swirl fan was used to spread the airborne *E. coli* in the chamber equally.

#### 2.1.5. Dust Monitoring

To monitor the dust concentration throughout the experiment, a dust concentration monitor (DustTrak DRX aerosol monitor 8533, TSI Inc., Shoreview, MN, USA) was used to measure the mass concentration of dust particles of different sizes [12]. DustTrak was capable of measuring dust particles of < 1.0 µm, 1.0–2.5 µm, 2.5–4.7 µm, 4.7–10.0 µm, and > 10.0 µm. The record intervals of DustTrak were 1 s. In 10 min of the experiment, a total of 600 data points were collected to monitor dust concentrations. The dust concentration and particle size were measured in this study, and the findings showed that the particle concentration was relatively consistent between experimental events [12].

#### 2.1.6. Air Samplers

An All-Glass Impinger (AGI-30, Ace Glass, Vineland, NJ, USA) and an Andersen six-stage impactor (Andersen impactor TE-10-800, Thermo Fisher Scientific, Inc., Franklin, MA, USA) were used in this study (Figure 1). The AGI-30 was proven to be an efficient sampler used for dry-aerosolization conditions [15]. The AGI-30 runs at a rate of 12.5 L min^−1^. The airborne compounds were pulled using a vacuum pump via a fine nozzle, where they were accelerated before impacting directly into the 20 mL TSB. The size distribution of airborne *E. coli* carried by poultry dust particles were monitored using the Andersen impactor. The sampler operates at 28.3 L min^−1^. It can separately collect airborne microorganisms of varied sizes of > 7.0 µm, 4.7–7.0 µm, 3.3–4.7 µm, 2.1–3.3 µm, 1.1–2.1 µm, 0.65–1.1 µm, respectively, from stages 1 to 6. The impactor separates dust particles (carrying *E. coli*) into seven different size ranges and collects them onto seven agar plates/stages via impaction mechanism. This impactor creates different air speeds. When dust particles in the air stream impact onto an agar plate at a speed, only particles above a certain size can be impacted on the plate. Smaller particles are transported by the air stream to the next stage where a higher air speed is created, allowing collection of smaller particles on another agar plate. Details regarding the cascade impaction mechanism were published in the paper by Andersen [17].

### 2.2. Experimental Design and Procedures

#### 2.2.1. System Design

A system was designed to simulate the conditions of the environment in the poultry houses. A sketch map of the system was shown in Figure 2. Two treatments were applied, one with UV lamps (one and two UV lamps) and the other without. Each system was made up of two chambers connected by an aluminum scrubber. The aerosolization system, which was described in 2.1.4., was used to aerosolize airborne *E. coli* attached to dust particles in the upstream chamber. To collect viable airborne *E. coli* generated from the aerosolization system, samplers were put in the upstream chamber. Two UV lamps (UVC lamp, Konideke, Yongchang, China) with a wavelength of 254 nm were installed in the aluminum scrubber. The two UV lamps are installed symmetrically in the scrubber and positioned on wall of scrubber (Figure 2). With one UV lamp positioned on wall of the scrubber, only one side of the airborne dust particles is exposed under UV irradiation. Symmetrically positioning two UV light bulbs on wall of the scrubber will increase the irradiation area on both sides of the dust particles, increasing the possibility of *E. coli* being exposed to UV rays. Another set of samplers was put in the downstream chamber to capture viable *E. coli* attached to dust particles after being irradiated with UV light. Air-in ports with high-efficiency particulate air (HEPA) filters were added to both chambers. The downstream chamber’s outflow was connected to a vacuum pump. The air filters served to keep airborne *E. coli* and dust particles out of the laboratory, while the vacuum pump helped to direct and control the airflow. The decrease rate was obtained by comparing the concentrations of airborne *E. coli* in the two chambers. In the system, the decrease of airborne *E. coli* was studied at varying air speeds (from 0.11 to 2.61 m s^−1^—typical airspeed range in the poultry houses [18]) and UV irradiance levels. Airborne *E. coli* may deposit on the surface of the test system, referred to as physical loss, which should be determined and excluded from the calculation of biological inactivation by UV light. The same operation was done in the testing system without the UV lamp, and the findings provided data on the physical deposition of airborne *E. coli* during air movement in the testing system. The physical loss was calculated by comparing the concentrations in the upstream and downstream chambers.

#### 2.2.2. System Setup and Sampling Collection

A total of 240 grams (240 g) of litter containing 8 log_10_ CFU (g litter)^−1^ of *E. coli* was produced and added to the mixer. To assist in equally spreading the dust particles carrying *E. coli*, the mixer was put in the center of the chamber. Suction cups were used to secure the mixer to the chamber surface, preventing it from sliding throughout the running process. The stir fan was positioned in the chamber’s corner to aid in the distribution of airborne particles. The test would take ten minutes to complete. The samplers, mixer, DustTrak, and fan were all turned on at the same time. The samplers’ sampling ports were adjusted to a set height of 27 cm. In addition, in each aerosolization event, the sampler positions were switched at random to reduce the location impact. The dust concentration in the two chambers was also measured using DustTrak. The temperature in the chamber was fixed at around 22.6 °C, with RH of ~60%.

#### 2.2.3. UV Light Intensity Distribution

A mathematical model, UVCalc software (UVCalc®, Bolton Photosciences Inc., Edmonton, AB, Canada) was used to simulate the UV light intensity distribution in the UV scrubber. The UVCalc software is widely used to support the design of UV reactor in the most accurate way [19]. However, because of the optical complexity of the scrubber, a UV light meter was used to validate the accuracy of the model. A UV light meter (Amtast USA Inc., Lakeland, FL, USA) was used to measure the UV light intensity. UV light in the range of 248 nm to 262 nm was measured by the UV meter. The measuring range for irradiance is 0.001 mW cm^−2^ to 39.99 mW cm^−2^. After installing the UV lamps in the aluminum scrubber, they were measured at various distances to obtain the most UV intensity distribution in the connection tube.

### 2.3. Calculation of E. coli Concentration and Inactivation Rates

#### 2.3.1. Determining Size Distribution of Airborne *E. coli* Carried by Poultry Dust Particles

The size distribution of airborne *E. coli* carried by poultry dust particles was monitored using an Andersen impactor. The Andersen impactor determines the counts of *E. coli* that carried by different (seven) size ranges of poultry dust particles. The Andersen impactor has six stages, each with one Petri dish. TSA was used to prepare each Petri dish. After being aerosolized, poultry dust particles carrying *E. coli* were sucked into the inlet of the Andersen impactor during the sampling process. The particles which carry *E. coli* then went through six stages of the sampler. TSA plates were used to capture dust particles carrying *E. coli* with sizes that corresponded to each stage. The *E. coli* on the agar plates were placed in an incubator for 24 h, at 37 °C and allowed *E. coli* to grow.

#### 2.3.2. Determining Airborne *E. coli* Concentration

AGI-30 was used to collect *E. coli* from the air (in TSB medium). With the use of a vacuum pump that was directly connected to AGI-30, airborne *E. coli* carried by dust particles were pulled into the intake of the AGI-30 and passed via a fine nozzle into the TSB solution. In the collection vessel, 20 mL of TSB medium was prepared. The total culturable airborne *E. coli* collected by the sampler was quantified using the traditional culture procedure. In the traditional culture procedure, each air sample (in liquid form) was utilized to quantify culturable *E. coli*. Total 0.1 mL serially diluted with the ratio of 1:10 samples were plated onto TSA agar plates after vortexing for 5 seconds. The plates were aerobically incubated for 24 h at 37 °C. On plates, the visible *E. coli* colonies (30 to 300 colonies) were counted. Airborne *E. coli* concentrations, in logarithm colony-forming units per cubic meter (log_10_ CFU m^−3^), were determined based on Equation (1).
(1)$$C=\log_{10}\left(\frac{N\;\times 10^{n}}{V_{p}}\times {\;V}_{s}\times \frac{1}{V_{a}}\right),$$
where C: the airborne bacteria concentration, log_10_ CFU m^−3^; N: the number of colonies on a countable plate (30 to 300 colonies); n: serial dilution factor (n = 0 for undiluted sample, n = 1 for 10-fold diluted sample, etc.); V_p_: the sample volume plated, mL (V_p_ = 0.1 mL in this study); V_S_: the total volume of the original liquid sample, mL; V_a_: the total air volume sampled using the bioaerosol samplers, m^3^.

#### 2.3.3. Inactivation Rates

The inactivation rate refers to the inactivation or the loss of airborne *E. coli* after passing through the UV light scrubber. The rate of the inactivation was calculated by Equation (2):
(2)$$\mathrm{Inactivation}\;\mathrm{rate}=(1-\frac{C2}{C1}\times \frac{1}{1-a})\times 100\%$$
where Inactivation rate: biological loss caused by the UV lamps, %; C2: the airborne bacteria concentration in the downstream chamber, CFU m^−3^; C1: the airborne bacteria concentration in the upstream chamber, CFU m^−3^; a: physical loss caused by the system, %.

The k-value was an additional metric used to represent how UV light affected microbiological survival [20]. The k-value is the inactivation rate of bacteria normalized by UV irradiance and contact time. The k-value was determined by using Equation (3):(3)$$k=-\frac{\log_{10}\left(\frac{C2}{C1}\times \frac{1}{1-a}\right)}{F},$$
where k: k-value, cm^2^ mJ^−1^; C2: the airborne bacteria concentration in the downstream chamber, CFU m^−3^; C1: the airborne bacteria concentration in the upstream chamber, CFU m^−3^; a: physical loss caused by the system, %; F: is the product of UV irradiance, mW cm^−2^, and the contact time (from 5.6 s to 0.23 s in this study).

#### 2.3.4. Reynolds Number

At high wind speeds, the flow of air in the system was affected by turbulent flow which can lead to deviations of the k-value. To verify whether the flow of air in the system was affected by turbulent flow, the Reynolds number which is an indicator for turbulent flow was calculated [21]. The Reynolds number in a pipe was calculated by the following Equation (4):(4)NRe=ρvdµ,
where N_Re_: Reynolds number; ρ: the density of the fluid, kg m^−3^; v: the flow speed, m s^−1^; d: the hydraulic diameter of the pipe, m; µ: the kinematic viscosity, kg m^−1^ s^−1^. In this system, since the airflow was circulated through the UV scrubber, thus, the hydraulic diameter is equal to the inside pipe diameter.

### 2.4. Statistical Analysis

The system was tested with three doses of UV light which were zero, one, and two UV lamps. With each dose of UV light, there were 4 wind speed levels being tested at 0.11, 0.51, 1.74, and 2.61 m s^−1^ corresponding to the contact times of 5.62, 1.17, 0.34, and 0.23 s. Temperature and RH were kept stable during experiments. With each wind speed level, the test was repeated three times which makes the total observations of 36 data points. The GLIMMix ANOVA model running on Statistical Analysis System (SAS 9.4, SAS Institute Inc., Cary, NC, USA) was used in statistical analysis to assess the inactivation rate of airborne *E. coli* and the k-values as influenced by the as influenced by the airborne *E. coli* and initial bacterial concentrations. The significant level was applied as the *p*-value of 5%.

## 3. Results

### 3.1. UV Light Intensity Distribution

Two UV lamps were positioned oppositely inside the tube. In a dusty environment, airborne *E. coli* can be carried by dust particles which prevent UV light from irradiating *E. coli*. Placing two symmetrical UV lamps can increase the UV irradiance exposure to *E. coli*. The UV light intensity distribution of one and two UV lamps simulated by UVCalc software is shown in Figure 3. The distributions of UV intensity were not uniform. With one UV lamp installation, the UV fluence rate decreased as the distance away from the UV lamps increased. The overall means of UV irradiations in the central plane were 3687 µW cm^−2^ and 7434 µW cm^−2^ for one UV lamp and two UV lamps, and in the total scrubber were 1707 µW cm^−2^ and 3422 µW cm^−2^ for one UV lamp and two UV lamps, respectively. The UV fluence rate measurement was validated by the UV light meter at different distances away from the UV lamps. A total of 12 points at different distances were measured for each UV lamp setup. The average fluence rate of the 12 points measurement were 4884 µW cm^−2^ and 9860 µW cm^−2^ for one and two UV lamps. The average fluence rate of 12 points with the corresponding distances calculated by the model is 5048 µW cm^−2^ and 10,869 µW cm^−2^. The relative accuracy of the measured data and the data taken from the model has a difference of about 3% and 9% for one and two UV lamps, respectively. This shows the reliability of the model as the UV light meter has an accuracy of ±5%. In two UV lamp setup, there is a difference of more than 5% compared to the model data. This can be explained by limitations in the measurement process. UV meter sensor can only cover a certain irradiance angle. Therefore, when measuring the UV intensity of two lamps symmetrically positioned on wall, it will not be able to cover the entire incident light, leading to a slight decrease in fluence rate. In one UV lamp setup, the UV meter sensor cover the irradiance angle better than in two UV lamp setup, making a better accuracy rate.

### 3.2. Size Distribution of E. coli Attached to Dust Particles and Dust Particles

The size distribution of airborne *E. coli* carried by poultry dust particles are shown in Table 1. In the upstream chamber, most *E. coli* were found in particles larger than 7 µm. The second sizable portion of *E. coli* was those attached to particles in the range of 4.7 to 7 µm. The least *E. coli* was found in particles smaller than 2.1 µm which accounted for the total culturable *E. coli*. 

The mass size distribution of dust particles was measured by DustTrak, and the results are shown in Figure 4. With a proportion of 62.3%, the majority of dust particles were less than 1 µm in size. The rest of the dust particles had size ranges of 1.0–2.5 µm, 2.5–4.7 µm, 4.7–10.0 µm, and larger than 10.0 µm, with proportions of 2.1%, 3.2%, 16.8%, and 15.7%, respectively. As shown in Table 1 and Figure 4, although most dust particles were smaller than 1 µm, the size distribution of bacteria attached to dust particles was mainly larger than 2.1 µm. This demonstrated that most airborne *E. coli* are associated with dust particles greater than 2.1 µm in size. It can be explained by shielding effect [22]. In a previous study [22], authors mentioned that the virus associated with bigger particles may be more protected from changes in the ambient environment than viruses that live as a singlet or bind to smaller particles. Thus, *E. coli* in this study may be shielded from environmental ambient when carried by large particles. Small particles had less of a protective impact on bacteria adhering to them. This could be one reason of the quick death of airborne *E. coli*.

### 3.3. Contact Time Effect on Airborne E. coli

The contact time affected the concentration of viable *E. coli*. As the wind speed increased, the contact time decreased. Additionally, when contact time decreased, the concentration of *E. coli* in the system also decreased as shown in Figure 5. This was consistent with the real situations in the poultry house. When wind speed increases, the concentration of dust particles and pathogens decreases [23].

### 3.4. Physical Loss of Testing System

The physical loss was calculated by comparing the concentrations (Appendix A) in the upstream and downstream chambers. In this system, the physical loss of *E. coli* was approximately 83% or 0.8 log_10_ reduction. The physical loss of the testing system is shown in Figure 6.

### 3.5. Inactivation Efficiency of UV Light

Temperatures and RH are shown in Table 2. The temperatures and RH remained stable over the experiments. However, there was a slight decrease in RH at the short contact time (0.23 s). An explanation could be that dehydration affected RH. At a short contact time or high wind speed, the dehydration effect in the air inside of chambers would increase leading to the decrease of RH [15]. 

The inactivation efficiencies of UV light are shown in Table 3. The concentrations of airborne *E. coli* (Appendix A) were reduced significantly for all treatments with UV lamps. The inactivation rates (biological loss), after removing the physical loss caused by the system, varied from 99.87% and 99.95% at 5.62 s of contact time with irradiance levels of 1707 µW cm^−2^ and 3422 µW cm^−2^ to 72.90% and 86.60% at 0.23 s of contact time with irradiance levels of 1707 µW cm^−2^ and 3422 µW cm^−2^. Results also showed that the inactivation rates decreased accordingly with the contact times. As wind speed increased from 0.11 to 2.61 m s^−1^, the time that airborne *E. coli* was exposed to light decreased from 5.6 s to 0.23 s. Therefore, the UV irradiance doses exposed to airborne *E. coli* also decreased, leading to a decrease in the inactivation rates.

The k-values are shown in Table 4. The k-values were not similar among different treatments. When the contact times and the number of UV lamps changed, the k-values changed accordingly. Typically, in the same bacteria strain, the k-value would be unchanged when they expose to the same disinfectant. However, in this study, the k-value varied when the contact time changed. An explanation was that the turbulent flow of high wind speed affected the k-value. To verify that, we calculated the Reynolds number which is an indicator for turbulent flow. The results showed that at the 5.62 s contact time, the Reynolds number was 1738 which was smaller than 2000 which means the laminar flow [21]. However, for 1.17, 0.34, and 0.23 s contact times, the Reynolds numbers were 8863, 29,370, and 44,576, respectively, which were significantly greater than 4000, indicating turbulent flow [21].

## 4. Discussion

The designed system was assessed for the effect of UV light on the inactivation of airborne *E. coli* carried by poultry dust particles. The inactivation rate was examined at different air speeds with the aid of a vacuum pump from 0.11 m s^−1^ to 2.61 m s^−1^ corresponding to the contact time from 5.62 s to 0.23 s, and different UV radiation intensity (1707 µW cm^−2^ and 3422 µW cm^−2^). Before conducting the experiment, to ensure the expected UV irradiation intensity, UVCalc software was applied to simulate the UV light intensity distribution in the UV scrubber. The maximum UV light irradiance was observed to be close to the UV lamps, indicating that the UV irradiance in the UV scrubber was not spread uniformly. UV light irradiance reached up to 24,759 µW cm^−2^ for one UV lamp or 25,864 µW cm^−2^ for two UV lamps when close to the lightbulb, and it gradually drops to 946 µW cm^−2^ for one UV lamp or 3450 µW cm^−2^ for two UV lamps when 24 cm away from the UV bulb for one UV lamp or 17 cm away from the UV bulbs for two UV lamps. The UV light irradiance was measured in laboratory conditions with a clear environment. In poultry house conditions, UV light irradiance may not reach such the level. Especially with the dusty conditions of the poultry environment, UV light bulbs can be covered with dust and reducing their ability to sterilize. Prior research found that the dust concentration might reach 81.33 mg m^−3^ in poultry houses [24]. When summer comes, the ventilation system must work harder resulting in increased airflow through the UV system. This would increase the amount of dust particles flowing through the UV tube, leading to a reduction of disinfectant ability.

In a past study [11], UV light has been extensively explored and was well recognized as a technique that can inactivate germs by preventing DNA replication. According to earlier research [11], UV radiation was particularly effective in killing *E. coli* in water, droplets, and surfaces used in the processing of food. The same was true for airborne *E. coli* carried by poultry litter particles. In general, a positive relationship between UV irradiance level and the inactivation rate was observed in this study. The positive relationship between UV irradiance and the inactivation rate was also reported in previous studies [20,25]. At the high wind speed (2.61 m s^−1^ or 0.23 s of contact time), the inactivation of the airborne *E. coli* drastically increased when more UV irradiances were applied (3422 µW cm^−2^ versus 1707 µW cm^−2^). While a single lamp (average of 1707 µW cm^−2^) killed 72.90 ± 2.57% (or 0.6 ± 0.0 log_10_ reduction) of the bacteria, two lamp (average of 3422 µW cm^−2^) inactivation rate up to 86.60 ± 1.35% (or 0.9 ± 0.1 log_10_ reduction) of the bacteria. When the number of UV lamps was raised from one to two at lower wind speeds (≤1.74 m s^−1^), a positive relationship was still seen, but the difference was not as obvious as it was at higher wind levels. At low wind speed, the exposure time of airborne *E. coli* to UV light increased significantly from 0.23 s (at 2.61 m s^−1^) to 5.62 s (at 0.11 m s^−1^), resulting in about 3 log_10_ of the bacteria being inactivated. As a result, it is difficult to observe the difference as clearly as at high wind levels where the exposure period was short. In addition, the results also pointed out that the inactivation of airborne *E. coli* was not linearly related to the UV irradiance used. A previous study [25] also reported a similar result. The study [25] reported that an increase in UV irradiation (within a similar contact time) above 5 µW cm^−2^ did not yield a proportional increase in inactivation rate. A possible explanation is that the dust particles which carry *E. coli* can block a certain amount of UV irradiation to *E. coli*. This can lead to a decrease in UV irradiation efficiency and makes the increase in irradiance not proportional to inactivation rates.

In a previous study [12], we examined the survivability of airborne *E. coli* carried by poultry dust particles in laboratory conditions. The airborne *E. coli* had a half-life time of over 5.7 minutes. The half-life time is the amount of time required for bacteria to decline by half or 50%. In this study, the survival time of the bacteria when exposed to UV light was much shorter. Specifically, the inactivation rate was 72.90 ± 2.57% with irradiance level of 1707 µW cm^−2^ or 86.60 ±1.35% with irradiance level of 3422 µW cm^−2^ at the 0.23 s contact time, and up to approximately 100% with irradiance levels of 1707 µW cm^−2^ and 3422 µW cm^−2^ at the 5.6 s contact time. At 1707 µW cm^−2^ and 3422 µW cm^−2^ UV irradiance levels, 99.87 ± 0.07% and 99.95% ± 0.04 (or 2.9 ± 0.3 log_10_ reduction and 3.5 ± 0.5 log_10_ reduction) of *E. coli* were eliminated in 5.6 s of contact time compared to only 50% in 5.7 min in the normal condition. Therefore, it can be affirmed that the use of UV light to reduce airborne *E. coli* carried by poultry dust particles was extremely effective under the experimental conditions.

Based on the obtained results, the k-value was calculated accordingly. In principle [20,26], the k-value should be the same for the same bacteria strain exposed to the same disinfectant. However, by the effect of turbulent flow generated at the high wind, the k-value was impacted. The results showed that, at the contact time of 5.62 s, the Reynolds number was smaller than 2000 which means the laminar flow [21]. In contrast, at shorter contact times (1.17, 0.34, and 0.23 s), the turbulent flow appeared. This turbulent flow is an unstable airflow and might affect the k-values. The bacteria in the laminar flow expose to just one site to UV radiation. On the contrary, when the airflow is unsteady, dust particles might spin around. As a result, it increases the chance that *E. coli* are exposed to UV radiation. The inactivation rate is affected by wind speed by two means. On one hand, higher wind speed reduces the contact time, which compromises the inactivation rates; on the other hand, higher wind speed increases turbulence that alters UV exposure by *E. coli* and thus inactivation rates. The latter is an interesting assumption that has never been reported by other studies and requires further research. In addition, when installed UV lamps varied, k-values also varied. One UV bulb may effectively eliminate microorganisms. However, doubling the number of UV lamps did not raise inactivation rates proportionally. The k-value is defined as the inactivation rates adjusted by irradiance and contact periods. So, since the inactivation rates did not rise according to the number of UV lamps, we may calculate the various k-values.

To apply the UV system on an industrial scale, contact time, wind speed of the ventilation system, UV irradiance level, and dust concentrations needs to be considered. In commercial poultry houses, the ventilation rates will vary depending on the season and variety of environmental conditions. Thus, when applying to the industrial scale, the varies in conditions may affect inactivation effectiveness. In addition, the poultry houses are typically dusty which can affect the UV system in the long-term run. The poultry dust can cover the UV lamp surfaces which reduces the UV irradiance, and thus, reduce the effect of the UV system. A periodic cleaning schedule is suggested when applying the system in poultry houses. Next, even though UVA (wavelength of 315–400 nm) was well studied and proven that it had positive effects on poultry [27], UVC (wavelength of 200–300 nm) was not well studied yet. Therefore, the application of UVC, in this case, 254 nm, into the poultry environment also needs to consider its impact on the poultry. Finally, study results suggested that one UV lamp was able to create the irradiance level of 1707 µW cm^−2^ which effectively (92.6% of inactivation rates) killed airborne *E. coli* generated in the system at the contact time of 0.34 s or longer. The upstream chamber of the system has a volume of 0.54 m^3^ with a concentration of 6.7 log_10_ CFU m^−3^ of airborne *E. coli* which is consistent with a previous study [12]. Given the typical size of a poultry house, it is necessary to install multiple UV light systems corresponding to the volume of the house to ensure complete coverage of the poultry house. In addition, an increasing number of UV lamps would not proportionally increase the inactivation rates. The poultry house, however, is a dusty environment where airborne dust particles can prevent some UV exposure to *E. coli*. Thus, it is necessary to increase the irradiation source, or in other words increase the number of UV lamps. The system would be installed before the outlet of ventilation system to reduce airborne *E. coli* emitted outside the poultry houses.

## 5. Conclusions

This study investigated the effect of UV light on the inactivation of airborne *E. coli* carried by poultry dust particles in laboratory conditions. The laboratory conditions remained stable at about 22.6 °C with an RH of 60%. In this study, a system that simulated the actual conditions of the poultry houses was designed to evaluate the inactivation efficiency. Based on the results, we conclude that (1) the inactivation rates reduced from approximately 99.87% and 99.95% at 5.62 s of contact time with 1707 µW cm^−2^ and 3422 µW cm^−2^ of irradiance levels to 72.90% and 86.60% at 0.23 s of contact time with 1707 µW cm^−2^ and 3422 µW cm^−2^ of irradiance levels; (2) the average of UV irradiation were 1707 µW cm^−2^ and 3422 µW cm^−2^ for one UV lamp and two UV lamps, respectively; (3) turbulent flow might affect the inactivation efficiency of the UV system. The results of this study will help to bring up an idea of an affordable mitigating system for airborne pathogens.

## Figures and Tables

**Figure 1 animals-12-03170-f001:**
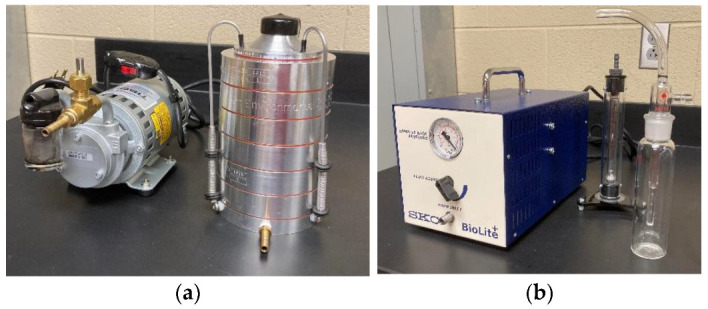
Two samplers: (**a**) Andersen six-stage impactor; (**b**) AGI-30 impinger.

**Figure 2 animals-12-03170-f002:**
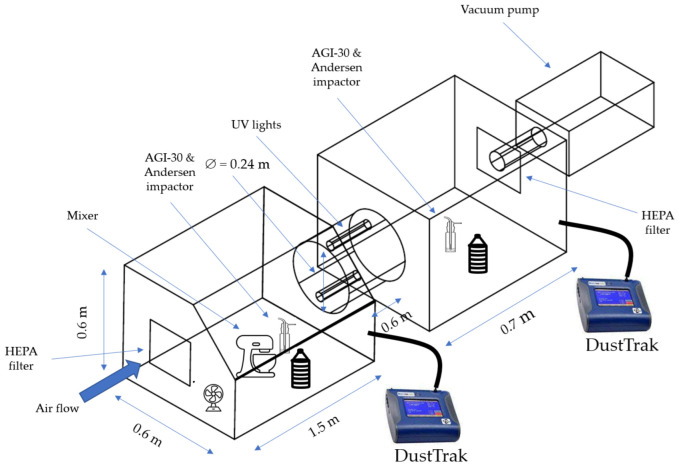
Testing system designed for examining the inactivation efficiency of UV light on airborne *E. coli*.

**Figure 3 animals-12-03170-f003:**
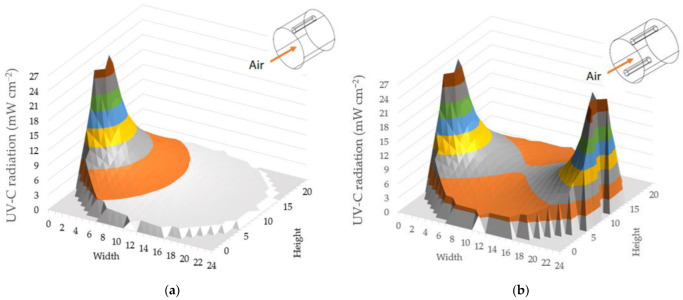
UV light intensity distribution of (**a**) one UV lamp; (**b**) two UV lamps.

**Figure 4 animals-12-03170-f004:**
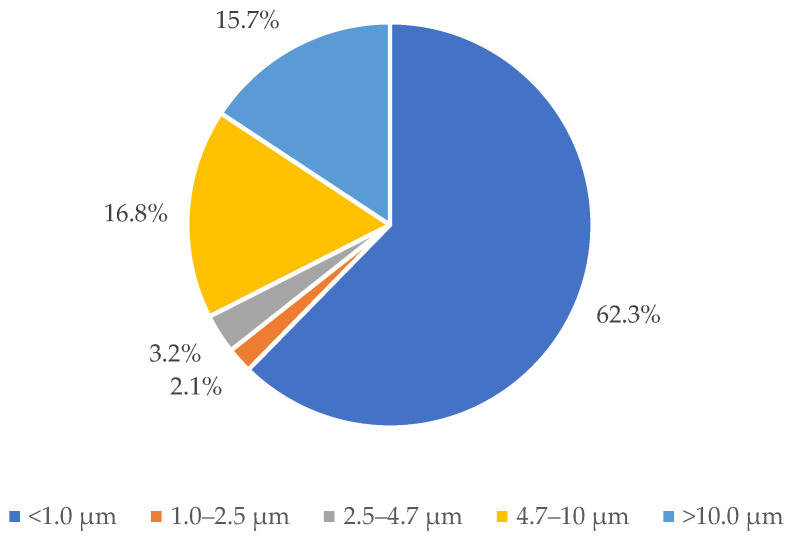
Size distribution of poultry dust particles.

**Figure 5 animals-12-03170-f005:**
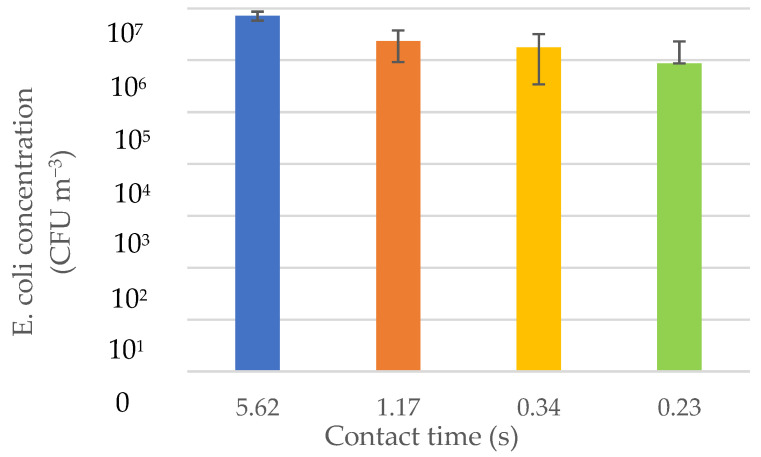
Contact time effect on *E. coli* concentration.

**Figure 6 animals-12-03170-f006:**
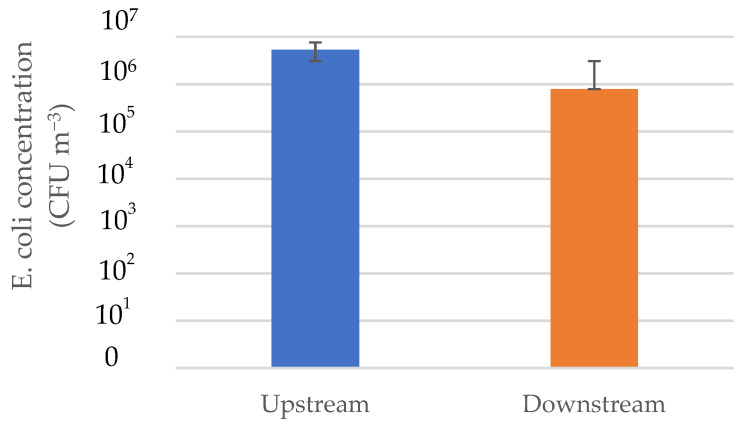
Physical loss of airborne *E. coli*.

**Table 1 animals-12-03170-t001:** Size distribution of airborne *E. coli* carried by poultry dust particles in upstream and downstream chambers.

Contact Time (s, Mean ± SD)	Chamber	>7.0 µm (%)	4.7–7.0 µm (%)	3.3–4.7 µm (%)	2.1–3.3 µm (%)	1.1–2.1 µm (%)	0.65–1.1 µm (%)
5.62 ± 0.91	Upstream	47.52	30.60	10.05	10.64	0.23	0.96
	Downstream	25.85	27.56	24.85	19.41	0.74	1.59
1.2 ± 0.06	Upstream	34.32	24.70	6.76	15.57	10.03	8.62
	Downstream	31.30	29.09	7.85	24.32	5.38	2.06
0.34 ± 0.01	Upstream	34.76	20.86	9.63	9.09	19.25	6.41
	Downstream	23.74	37.40	10.78	7.98	15.88	4.22
0.23 ± 0.01	Upstream	42.75	23.64	5.72	8.86	15.25	3.78
	Downstream	62.30	13.09	10.30	5.52	4.80	3.96

**Table 2 animals-12-03170-t002:** Temperature and relative humidity correspond to contact times and the number of UV lamps.

Contact Times (s, Mean ± SD)	Number of UV Lamps	Temperature (°C, Mean ± SD)	Relative Humidity (%, Mean ± SD)
5.62 ± 0.91	1	23.0 ± 0.7 ^a^	61 ± 7 ^a^
	2	23.0 ± 0.7 ^a^	61 ± 7 ^a^
1.2 ± 0.06	1	22.0 ± 1.4 ^a^	60 ± 2 ^a^
	2	22.0 ± 1.4 ^a^	60 ± 2 ^a^
0.34 ± 0.01	1	22.5 ± 0.5 ^a^	61 ± 3 ^a^
	2	22.5 ± 0.5 ^a^	61 ± 3 ^a^
0.23 ± 0.01	1	22.7 ± 1.3 ^a^	56 ± 5 ^b^
	2	22.7 ± 1.3 ^a^	56 ± 5 ^b^

Note: ^a^, ^b^ mean in the same column with different letters are different (*p* < 0.05). SD means standard deviation.

**Table 3 animals-12-03170-t003:** Inactivation rates of UV light correspond to contact times and the number of UV lamps.

Contact Times (s, Mean ± SD)	Number of UV Lamps	Inactivation Rates (%, Mean ± SD)	Log Reduction (log_10_)
5.62 ± 0.91	1	99.87 ± 0.07 ^a^	2.9 ± 0.3
	2	99.95 ± 0.04 ^a^	3.5 ± 0.5
1.2 ± 0.06	1	93.97 ± 0.36 ^b^	1.2 ± 0.0
	2	96.85 ± 1.23 ^c^	1.6 ± 0.2
0.34 ± 0.01	1	92.60 ± 0.63 ^d^	1.1 ± 0.0
	2	95.40 ± 0.59 ^e^	1.3 ± 0.1
0.23 ± 0.01	1	72.90 ± 2.57 ^f^	0.6 ± 0.0
	2	86.60 ± 1.35 ^g^	0.9 ± 0.1

Note: Means with the same letter are not significant different (*p* < 0.05). SD means standard deviation.

**Table 4 animals-12-03170-t004:** K-values correspond to contact times and the number of UV lamps.

Contact Times (s, Mean ± SD)	Number of UV Lamps	k-Values (cm^2^ mJ^−1^, Mean ± SD)
5.62 ± 0.91	1	0.300 ± 0.106 ^a^
	2	0.171 ± 0.059 ^ab^
1.2 ± 0.06	1	0.608 ± 0.145 ^c^
	2	0.378 ± 0.065 ^a^
0.34 ± 0.01	1	1.928 ± 0.35 ^d^
	2	1.136 ± 0.16 ^e^
0.23 ± 0.01	1	0.144 ± 0.072 ^b^
	2	0.114 ± 0.080 ^b^

Note: Means with the same letter are not significant different (*p* < 0.05). SD means standard deviation.

## Data Availability

Not applicable.

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
