# Peer review of "Effect of Ultraviolet Radiation on Reducing Airborne Escherichia coli Carried by Poultry Litter Particles"

_animals, 2022, doi:10.3390/ani12223170_

Round 1
Reviewer 1 Report
OVERVIEW
The manuscript details a bench-scale treatment efficacy study using a factorial design with 3 UV designs (2 lamps, 1 lamp, none), and 4 air flow rates. The study shows that UV irradiation can reduce airborne APEC concentrations by 1-3 log.
MAJOR COMMENTS
While I don't think it affects the potential for the study to be published, the reactor design is not optimal. In a cylindrical reactor, one lamp should be positioned in the center of the cross-sectional profile. Why was this not done/possible? A two-lamp cylindrical reactor with the lamps positioned on the wall is inefficient, as the dosage will be lowest in the center where the longitudinal airflow is greatest. UV reactor design can easily be optimized using software (e.g., http://www.uvcalc.com/).
I don't think the reported average UV doses are correct as they seem to be based on one poorly explained line of measurements across the profile of the reactor. The average across this line is not representative of the average across the whole circular profile, nor is the average across one circular profile representative of the average longitudinally along the reactor. It is not well explained where the lamps are positioned or where the measurements were taken.
The need for improved consideration of well known theory for reactor design is illustrated by the use of exponential and parabolic trendlines in Figure 3. Light intensity diminishes as a function of the square of distance from each source (for point sources without reflection). It should be possible to compare the measurements to a theoretical model. The Figure 3b data are puzzlingly asymmetrical about the midpoint (12 cm), suggesting some sort of error in the distance measurements or plotting.
MINOR COMMENTS
Please consider HuiYong et al. (2008), which addresses airborne E. coli concentrations in, upwind, and downwind of chicken barns - https://doi.org/10.1007/s11427-008-0021-0
Lines 59-61 suggest that inhalation of aerosolized APEC poses a greater threat to the poultry than the fecal-oral route or that settled aerosols outnumber more direct fecal contamination on surfaces. Please justify that inhalation is a significant transmission pathway and comment on the relative significance of airborne contamination and direct fecal contamination.
Is there an occupational health concern via inhalation for workers in these barns, or is the concern only about the health of the poultry?
Lines 92, 209, 217, 321 - Italics needed
Line 205 - Please check units (seems fishy)
Section 3.2 - The wording of this section (and elsewhere) is confusing so that it is unclear which two of the following are measured: 1) the size distribution of E. coli cells, 2) the size distribution of E. coli-bearing dust particles, 3) the size distribution of all dust particles.
Table 3 - I'd like to see a scatter plot of raw data with two series (2 lamps, 1 lamp), a y-axis showing log-reduction, and an x-axis showing windspeed.
Lines 369-370 - This is not an observation...it is basic engineering/physics that the UV intensity distribution will not be uniform, which is why reactor design is important.
Line 394 - Avoid language such as "almost 100% of the bacteria being inactivated". Log-reduction should be used, and can easily exceed 6-log in filtered water.
Reviewer 2 Report
The submitted research article entitled “Effect of Ultraviolet Radiation on Reducing Escherichia coli carried by Poultry Litter Particles” focuses on the development and testing of a laboratory model of a UV radiation-based approach for the reduction of Escherichia coli on aerosolized litter particles in poultry houses. According to the authors, this research can assist in the design of affordable mitigation systems for poultry houses. This field of research is extremely relevant for the poultry production sector and additional information regarding the optimization of biosecurity measures is valuable.
Despite the merit of the research presented, there are some issues regarding the manuscript that should be revised in order to improve the overall quality of the presentation and the interpretation of the results.
Introduction:
Please revise the references present in the introduction section and consider changing to more suitable ones.
Lines 44 – 45 – The work of Yu et al., 2016, does not seem to mention Avian Pathogenic Escherichia coli but does refer to Gram-negative bacteria.
Lines 55 – 56 – Zhao et al., seems to be focused on Highly pathogenic Avian Influenza rather than Avian Pathogenic Escherichia coli.
Lines 57 – 64 – It is not clear if the major concern is the possibility of aerosolized particles of poultry litter carrying Escherichia coli coming from the outside an entering the poultry houses, or the circulation of such particles within the houses or even the possibility of these particles leaving a poultry house and entering another one nearby, or all of the above. Nevertheless, for readers who are unfamiliar with poultry farming the authors could mention how should poultry litter be treated after usage.
Line 82 – The wind speed is presented in feet per minute. Please consider converting it to metric system.
Materials and Methods:
Line 183 - The wind speed is presented in feet per minute. Please consider converting it to metric system.
Results:
Line 272 - 275 – Please consider moving these sentences to the M&M section.
Line 275 – “The UV irradiance will be…”. Please correct accordingly.
Line 277 – 278 – The authors state that UV irradiation decreased as the distance from the UV lamps increased, though in Figure 3b that does not seem to be correct, since the lowest UV intensity presented was at a distance of 12 cm and the intensity rises again. Please comment and revise writing.
Table 1 – What is the meaning of the different superscript numbers? Please add to the manuscript that information.
Lines 308 – 310 – Please consider moving these sentences to the Discussion section.
Lines 314 – 316 – Please consider moving this sentence to the M&M section.
Lines 318 -319 – Does this mean that only 17% of all Escherichia coli added to the litter in the beginning of the experiment and aerosolized were subjected to UV light treatment?
Lines 324-326 – Please consider moving these sentences to the Discussion section.
Tables 2, 3 and 4 – There are no superscript letters on the tables. Please revise.
Lines 348 – 354 – Please consider moving these sentences to the Discussion section.
Discussion:
Line 374 – Please consider a synonym for the word disinfect in this scenario.
Lines 379 – 382 – Please consider rewriting of this section as it is almost the same as in lines 66-69.
Line 402 – “The airborne E. coli were reported that…”. Consider revising sentence.
Lines 410 – 412 – “…was extremely effective.” The authors should consider adding “under the experiment conditions”.
Line 415 – Should UV light be considered a disinfectant?
Lines 420 – 422 – The authors affirm that unsteady airflow increases the chance of UV light exposure. Nevertheless, the results presented point that increased wind speeds, and therefore reduced contact or exposure times and turbulent flow, are associated with reduced inactivation rates. Please comment.
Lines 446 – 447 – Repetition (lines 431 – 432). Consider revising sentence.
Lines 448 – 449 – The authors affirm that “it is necessary to increase the irradiation source, or in other words increase the num of UV lamps”, though in lines 423-426 the authors wrote “doubling the number of UV lamps did not raise the inactivation rates proportionally”. Please comment.
Additional comments/questions:
1 – The authors do not seem to mention or discuss why the smaller particles were not associated with Escherichia coli carriage, even though the majority of particles were <1,0 µm.
2 – The authors do not discuss possible applications of such mitigation systems to other pathogens.
3 – Though not discussed in to depth, it seems that the major factor influencing the results is wind speed.
4 – Where in the poultry houses should such devices be installed to effectively mitigate the problem posed by Avian Pathogenic Escherichia coli?
5 – What is the cost-benefit ratio of installing such devices in poultry houses when compared with adequate ventilation systems, including filters, and natural lighting?
Round 2
Reviewer 1 Report
The revised manuscript addresses many of my earlier comments well, and my remarks below are easily addressed.
MAIN COMMENTS
There continues to be a lack of clarity about what exactly the Andersen impactor is measuring, flip-flopping back and forth between sizes of E. coli and sizes of particles with E. coli. The suggestion that an E. coli can be greater than 7 microns in size seems rather dubious. A list of lines requiring polishing includes 255, 453, 454, 476, 483,
I tried a quick web search to find out what exactly this device measures and how it does so, and did not quickly find answers (the manufacturer's website was not useful). I suggest a sentence describing how the technology works to separate particles with bacteria from ones without and provide some references. I'm also curious about what results would be obtained if the test were run with dust particles including no E. coli as a type of control (though likely beyond the scope of this manuscript).
Please provide raw plate count data in supplementary content.
The discussion presents the UV intensity data as though these are surprising findings rather than foreknown patterns for the design employed.
MINOR COMMENTS
Line 2 - Airborne should not be italicized
Line 10 - "was" implies that it is proven as part of this study. Replace with "has been"
Line 21 - "ultraviolet UV (UV)"...delete the UV not in brackets
Line 32 - there is a "6" that needs to be deleted
Line 75 - "a predisposition of reproductive organs" may be missing "infection of" or something similar
Line 92 - "ultraviolet UVC (UVC)"...change to ultraviolet-c
Line 248 - manufacturer details are needed. For the Andersen impactor, specify a model number and if it is viable/non-viable.
Lines 292-293, 796 - Italics needed
Lines 293-295 - The added explanatory sentence isn't really helpful to me...more detailed explanation may be needed
Line 357 - rewording needed
Equations 2&3 - C1 and C2 are in cfu/(m^-3), not log cfu/(m^-3) as indicated. The units of "a" (fractional) are not provided.
Line 408 - "as influenced by the microbial species"...I thought there was only one (ATCC 25922)
Lines 458-459 - this is only true for the third contact time and to a lesser extent the first.
Line 485 - "2.1 m"...correct units
Line 490 - This seems to be a case of misstated causation.
Tables 2&3 - please note what type of test was used
Table 3 - The mean and standard deviation of log-transformed variables are not practically meaningful (though they are commonly reported in the literature). It isn't necessary to repeat tests on transformed data that were already performed on the untransformed data.
Line 715 - Cross-reference Table 3
Lines 727-728 - Could this arise from shielding?
Line 780 - Were all downstream samples non-detects? 100% implies that there were categorically no E. coli detected downstream.
